# Preparation and Application of ZIF-8 Thin Layers

Martin Schernikau [1,*], Jakob Sablowski [2,*], Ignacio Guillermo Gonzalez Martinez [1], Simon Unz [2], Stefan Kaskel [3] and Daria Mikhailova [1]

[1] Leibniz Institute for Solid State and Materials Research (IFW) Dresden e.V., Helmholtzstraße 20, D-01069 Dresden, Germany; i.g.gonzales.martinez@ifw-dresden.de (I.G.G.M.); d.mikhailova@ifw-dresden.de (D.M.)

[2] Institute of Process Engineering and Environmental Technology, Technische Universität Dresden, George-Bähr-Straße 3b, D-01062 Dresden, Germany; simon.unz@tu-dresden.de

[3] Department of Inorganic Chemistry, Technische Universität Dresden, Bergstraße 66, D-01069 Dresden, Germany; stefan.kaskel@tu-dresden.de

* Correspondence: m.schernikau@ifw-dresden.de (M.S.); jakob.sablowski@tu-dresden.de (J.S.)

**Abstract:** Herein we compare various preparation methods for thin ZIF-8 layers on a Cu substrate for application as a host material for omniphobic lubricant-infused surfaces. Such omniphobic surfaces can be used in thermal engineering applications, for example to achieve dropwise condensation or anti-fouling and anti-icing surface properties. For these applications, a thin, conformal, homogeneous, mechanically and chemically stable coating is essential. In this study, thin ZIF-8 layers were deposited on a Cu substrate by different routes, such as (i) electrochemical anodic deposition on a Zn-covered Cu substrate, (ii) doctor blade technique for preparation of a composite layer containing PVDF binder and ZIF-8, as well as (iii) doctor blade technique for preparation of a two-layer composite on the Cu substrate containing a PVDF-film and a ZIF-8 layer. The morphology and topography of the coatings were compared by using profilometry, XRD, SEM and TEM techniques. After infusion with a perfluorinated oil, the wettability of the surfaces was assessed by contact angle measurements, and advantages of each preparation method were discussed.

**Keywords:** metal–organic framework (MOF); electrochemical synthesis; anodic dissolution; liquid infused porous surfaces (LIPS); omniphobic surface; contact angle

## 1. Introduction

Omniphobic (hydrophobic and lipophobic) surfaces have a wide spectrum of applications, such as anti-fouling [1–3], anti-icing [4,5] or self-cleaning [6,7], in various fields, such as automotive, oil/gas, marine, paper and textile industries [8]. The fabrication of such surfaces is a current subject of research [9–11]. There are various approaches for the creation of omniphobic surfaces, applicable for a wide range of materials. A common approach is to achieve the so-called Cassie-Baxter state [8], where the droplet sits on top of a surface structure with trapped air underneath, so the liquid does not completely wet the surface. With this kind of surface, very high contact angles (greater than 150°) and very low contact angle hysteresis (lower than 5°) can be achieved for a wide range of fluids [12,13]. However, these surfaces are very sensitive to condensation and penetration effects. In this case, the air pockets underneath the droplets are flooded with the fluid, which leads to a transition into the Wenzel state [14,15] as shown in Figure 1. This Wenzel state creates much lower contact angles, often leading to a complete wetting of the surface by liquids with a low surface tension. Therefore, these surfaces are not suitable for applications in which high contact angles have to be maintained under pressure or during condensation.

Omniphobic lubricant-infused surfaces are resistant to this flooding problem, which is crucial for applications such as highly efficient dropwise condensation in heat exchangers [16].

On these surfaces, an omniphobic (perfluorated) oil with a low surface energy is held within a thin layer of the host material, which keeps the oil in place by using capillary forces [17]. This idea was inspired by nature and was introduced for the first time in 2011 [6,17]. The resulting surfaces have shown high contact angles with a small contact angle hysteresis (difference between advancing and receding contact angles) in additional to a self-healing effect and a better optical transmission behavior [17].

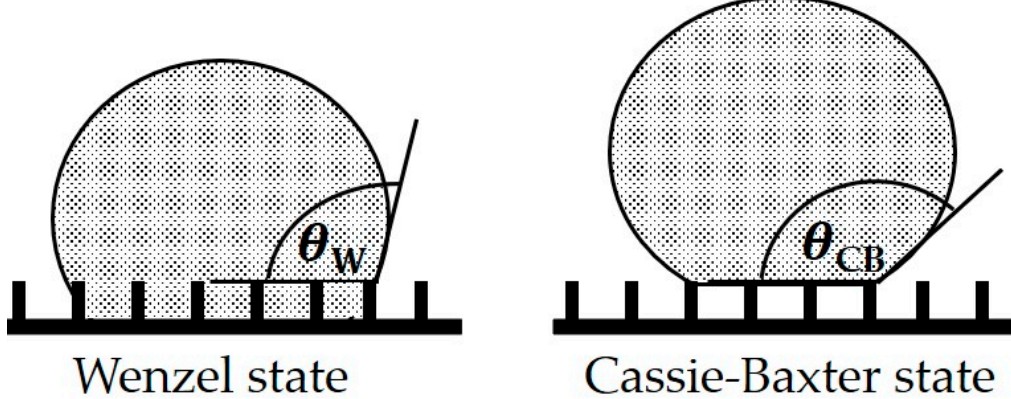

**Figure 1.** Different contact angle cases for rough homogeneous surfaces: Wenzel state and Cassie-Baxter state.

For industrial purposes, a stable, thin and convenient applicable coating for complex shaped components is required. In case of liquid-infused porous surfaces (LIPS), the supportive host material is the key to those desired properties.

Metal organic frameworks (MOFs) are known for their high specific surface area and porosity. They consist of metal ions or metal-oxygen-cluster, so-called "inorganic building units" (IBU), which are coordinated to organic ligands (or linkers) [18]. These IBUs and linkers create a coordination network with potential voids [19]. These voids can differ in size (depending on the linker) and often connect to each other, creating capillaries or pore networks [20]. In addition to pore size, MOFs can also easily be adjusted in their hierarchical structure, by using a huge variety of different templating methods [20], to control not only the pore size but also the macroscopic shape. Because of the high potential in customizability and a lot of possible applications, such as gas storage and separation as well as water absorption, drug delivery and catalysis, a lot of different synthesis routes are applied, such as solvothermal, mechanochemical, sonochemical and microwave-assisted synthesis as well as electrochemical synthesis.

By pretesting MOFs for their chemical and thermal resistance as well as their corresponding contact angles in the interaction with specific liquids, we decided to use ZIF-8 for our further investigations. ZIF-8 is a zeolitic imidazolate framework (ZIF) consisting of tetrahedrally coordinated $Zn^{2+}$ ions connected by 2-methylimidazolate linkers, creating an angle of 145° between these components, similar to the Si-O-Si angle in zeolites [21], as shown in Figure 2. ZIF-8 has been extensively studied and is highly suitable for creating a porous, thermally and chemically stable host layer [22–24] in LIPS, as well as it can be useful as thin layer in other applications such as catalysts [25] or sensors [26].

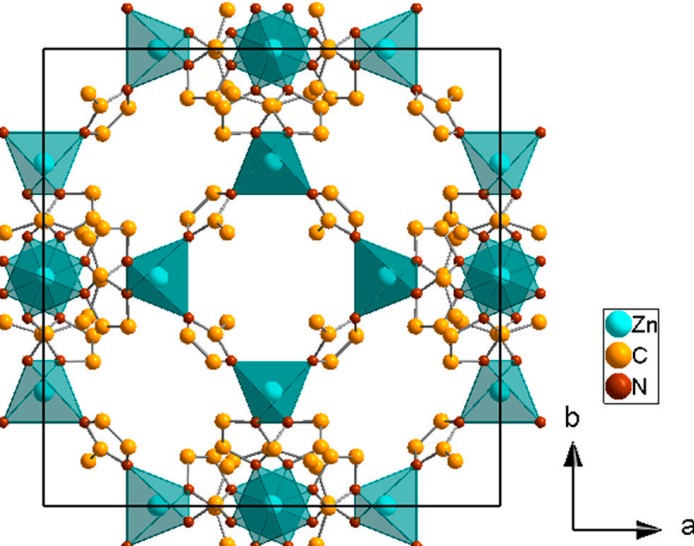

**Figure 2.** Crystal Structure of ZIF-8 (space group I-43 m, a = 16.8815(5) Å, V = 4811.0(4) Å, Z = 12) [27].

The electrochemical synthesis of an MOF powder was first developed and pioneered by the chemical company BASF [28]. In this synthesis, a metal electrode is used as metal source for the electrochemical oxidization of ions. The formed metal ions build the corresponding organometallic network with the dissolved linker. The rapid synthesis creates a good route for the preparation of powdered materials. However, a thin layer deposition of MOFs using an electrochemical route has been rarely explored so far. For example, some attempts were undertaken to achieve electrochemical layer deposition for MOFs [29], such as Cu-BTC framework HKUST-1 [30] and ZIF-8 [31]. Especially by using electrochemical techniques, formation of a thin, homogeneous and porous layer of MOFs on a surface may represent a very promising approach to fabricate stable, hierarchically superstructured surfaces of a complex shape for liquid infusion.

There are two common ways of electrochemical MOF synthesis. One represents the cathodic deposition [32], where the electrolyte contains a metal salt, the organic linker as well as a pro-base. The reduction of the pro-base generates a base, thus increasing the pH near the cathode. Then, the deprotonation of the organic linker is induced, which further coordinates the metal ions, and MOF precipitation occurs. Although this method is independent of the substrate used, it is not considered further in this work, since the method allows only a minor control of crystals formation during synthesis.

The anodic electrochemical route was introduced by Mueller and co-workers [28]. The metal ions of the electrode are released into the electrolyte when an appropriate voltage is applied. Then, they become coordinated with the organic linker form the electrolyte and precipitate on the electrode, forming a thin MOF layer. The major advantage of this method, apart from high mass transfer and application in flow reactors, is the high influence on the resulting MOF properties by changing synthesis conditions.

The use of an MOF as host material for LIPS was first introduced in 2018, via electrodeposition of Cu-BTC HKUST-1 [33]. In this work, an electrochemically deposited layer of HKUST-1 was used as supportive host. This MOF-layer was then infused with a per-fluorated oil resulting in high contact angles for polar, non-polar and low-surface-tension liquids (114° for water, 95 ° for diiodomethane and 61° for acetone). Since HKUST-1 shows larger contact angles and is probably more sensitive to water due to its components, we chose ZIF-8 for our studies.

Besides the resistance of penetration and flooding, omniphobic lubricant-infused surfaces show a small contact angle hysteresis. This can help to achieve dropwise condensation of low surface tension fluids, even if the contact angle is lower than 90° [34].

Since we want to test the applicability of these coatings for the further development of heat exchangers in future studies, copper is used as the substrate material for all coatings, because it has a particularly high thermal conductivity and is a typical material for heat exchangers. In this work, we report wetting properties of omniphobic lubricant-infused surfaces, created by the deposited host material. We compare electrochemically deposited ZIF-8 layers with two different types of ZIF-8/PVDF composite layers and discuss advantages of each preparation.

## 2. Materials and Methods

### 2.1. Electrochemical Anodic Deposition

To create an electrochemical deposited ZIF-8 coating on a copper substrate, the copper substrate was first galvanized with zinc. For this, the substrate was placed in an electrochemical cell, with a zinc counter electrode. The electrolyte contained 0.25 M sodium citrate ($Na_3C_6H_5O_7$*2 $H_2O$, Merck, Darmstadt, Germany, pure for analysis) and 0.2 M zinc sulfate ($ZnSO_4$*7 $H_2O$, Merck, Darmstadt, Germany, pure for analysis). The zinc coating was created by applying a potential of 1.7 V (vs. Mercury-mercurous sulfate electrode, MSE) over 10 min. The sample was washed with water three times. In the next step, the galvanized sample was placed in an electrochemical cell, with a platinum counter electrode, and a potential of −1.1 V (vs. MSE) was applied over 30 min. The electrolyte contained 1 M 2-methylimidazole (99% Acros Organics, Fair Lawn, NJ, USA) and 0.1 M sodium sulfate ($Na_2SO_4$*10 $H_2O$, Merck, Darmstadt, Germany, pure for analysis). In some experiments, ethanol solution (96%, Sigma-Aldrich, St. Louis, MO, USA) was added to the electrolyte.

### 2.2. Doctor Blade Technique for PVDF Layer

For creating a thin layer of PVDF (Polyvinylidenfluoride, Solef 21216, Solvay, Hannover, Germany) on the Cu foil, we added 40 mg of PVDF-powder into 800 μL 1-Methyl-2-pyrrolidone [NMP] (99%, Sigma Aldrich, St. Louis, MO, USA) and mixed them in a ball mill for 30 min at 25 Hz. The resulting dispersion was added to a doctor blade coating machine. The machine was set to 20 μm thickness and a forward speed of 5 mm/s. The layer was then dried overnight at room temperature and for an additional 24 h in a drying chamber at 80 °C.

### 2.3. Doctor Blade Technique for Composite Layer Containing PVDF Binder and ZIF-8

For creating a thin layer containing PVDF binder (Solef 21216, Solvay) and ZIF-8 (Basolite® Z1200, Sigma Aldrich, St. Louis, MO, USA) on the Cu foil, we added 40 mg of PVDF and 160 mg of ZIF-8 into 800 μL 1-Methyl-2-pyrrolidone [NMP] (99%, Sigma Aldrich, St. Louis, MO, USA). The components were mixed in a ball mill for 30 min at 25 Hz (8 steel balls, 3 mm in diameter were used). Approximately 500 μL of the resulting dispersion was added to a doctor blade coating machine; the steel balls were removed mechanically. The blade was set to 20 μm thickness and a forward speed of 5 mm/s. The resulting layer, with an average coating area of 100 cm$^2$, was than dried overnight at room temperature and for an additional 24 h in a drying chamber at 80 °C. Circles with a diameter of 20 mm were then punched out for later use and better manageability.

The same procedure was done for other mixing combinations of PVDF/ZIF-8 (20 mg/180 mg, 60 mg/140 mg and 80 mg/120 mg). The changed viscosity of the resulting suspension was roughly corrected by adding an additional 50 μL of NMP.

### 2.4. Successive Two-Layer Composite, Containing a PVDF-Film and a ZIF-8 Film

For creating a successive two-layer composite film, two mixtures were used. In total, 48 mg of PVDF were dissolved in 800 μL of NMP and 200 mg of MOF were added to 1 mL NMP. Both liquids were separately mixed in a ball mill at 25 Hz for 30 min (8 steel balls, 3 mm in diameter were used). The PVDF solution was then placed first on the coating machine and a 20 μm thin layer was created. After temporarily drying for 30 s, a second layer was added onto the PVDF coating. Subsequently, the MOF containing dispersion was

added onto the PVDF coating and the doctor blade machine was set to 35 μm thickness and the same forward speed of 5 mm/s. The resulting thickness of the outer MOF-coating was 15 μm. Circles with a diameter of 20 mm were then punched out for later use and better manageability. To produce lubricant-infused surfaces, the host material was infused with Krytox GPL 100 or Krytox GPL 105 (DuPont, Wilmington, DE, USA) after drying in a vacuum oven at 120 °C, overnight. For oil infusion, the sample was tilted by 45° and a droplet was set on the samples surface.

### 2.5. Contact Angle Measurements

Contact angles on the samples were measured using a contact angle goniometer (Krüss model DSA100 for droplet dosing and image capturing, Krüss (Hamburg, Germany) Advance software (Version 1.10), for profile detection and drop shape analysis). Contact angles were measured with deionized water and diiodomethane (99%, Alfa Aesar, Karlsruhe, Germany) in air at temperatures ranging from 20 to 25 °C. Advancing and receding contact angles were measured using the needle-syringe method. Droplets with a volume of 3 μL were carefully placed on the sample. To measure the advancing contact angle, the volume of the sessile droplet was slowly increased by pumping additional liquid through the needle into the droplet at a rate of 0.68 μL/s. The droplets reached a maximum volume of 20 μL. To measure the receding contact angle, the process was reversed and the volume of the sessile droplet was reduced. In both cases, the contact angle was measured as the contact line of the droplet started to move on the surface. In case of pinned droplets, there was no significant movement of the contact line as liquid was withdrawn from the droplets and no receding contact angle is reported. For the drop shape analysis, the conic section fitting method or the polynomial fitting method were chosen. At least three different locations on each surface were used to repeat the measurements and at least three measurements were conducted for each drop. Measured contact angle values are reported as averages with the associated standard deviation.

### 2.6. Further Methods

#### 2.6.1. SEM

SEM images were measured using a Zeiss (Oberkochen, Germany) Leo Gemini 1530. ZIF-8/PVDF composite layers were additionally coated with 20 nm carbon due to the low electrical conductivity.

#### 2.6.2. TEM

The TEM observations were performed with a FEI (Hillsboro, OR, USA) TECNAI F30 microscope operated at an acceleration voltage of 300 kV. The composite films were pulverized and transferred onto Cu/lacey C TEM grids (Plano) by direct contact.

#### 2.6.3. XRD

Phase analysis of materials after synthesis was carried out at room temperature using X-ray powder diffraction (XPD) with a STOE (Darmstadt, Germany) STADI P diffractometer (Cu–Kα1 radiation, λ = 1.54056 Å) in a transmission mode.

#### 2.6.4. Optical Profilometry

The measurements of the roughness parameters were carried out with a Micro Prof optical profilometer from Fries Research & Technology (FRT), Bergisch Gladbach, Germany. The measuring range of the measuring head extended to 600 μm. The two programs Acquire (V5.10.4.0, 2019) and Mark III (V3.9.25, 2013) from the same company were used for recording and processing. The surfaces were measured in a square with an edge length of 2 mm at an area resolution of 3 μm and roughness analyses were then determined within this area in a square with an edge length of 0.75 mm. The position of the square was varied at least three times.

## 3. Results

### 3.1. Electrochemical Thin Layer Deposition and Effect of Synthesis Conditions

Electrochemical synthesis of MOF materials represents a method to create a thin layer coating onto complex shaped surfaces in a fast and scalable way.

We applied the anodic method to create a thin layer coating of ZIF-8, in which zinc ions are electrochemically dissolved in the electrolyte, containing 2-methylimidazole as the linker. The two educts then form the resulting ZIF-8 network that is deposited on the substrate, because the concentration of zinc ions is significantly higher close to the surface of the working electrode. Since ZIF-8 must be deposited on the Cu substrate, the first step included deposition of metallic Zn on the Cu substrate, followed by the formation of ZIF-8. To examine the effect of different synthesis parameters for the anodic electrodeposition of ZIF-8, we changed individual parameters, based on a standardized synthesis process reported in Section 2.1. This standard synthesis is based on previous publications [31] and subsequent optimization for our application. The following parameters were gradually changed: linker concentration, potential, synthesis time and electrolyte composition. The resulting coatings were pre-evaluated for a homogeneous, stable and flat coating and then characterized by XRD and SEM for chemical purity and particle shape, as well as detailed information about microscopic coating quality. The electrochemical setup for the synthesis with a Cu plate as the working electrode is shown in Figure S3.

The deposited ZIF-8 particles of the standardized procedure were scratched off from the surface, and the resulting powder was measured with X-ray diffraction. The measured pattern fits well with the corresponding reflections of the ZIF-8 reference [35], as shown Figure 3. However, an additional reflection was observed at around 8.5°, corresponding to triclinic $Cu_{2.5}SO_4(OH)_3*2H_2O$ [36]. Most probably, during anodic deposition of ZIF-8, some bare Cu surface was created upon Zn dissolving, and reacted with electrolyte species. Note that the background of the measured diffractogram is quite high, especially in the starting region, indicating the presence of some amorphous components in the sample.

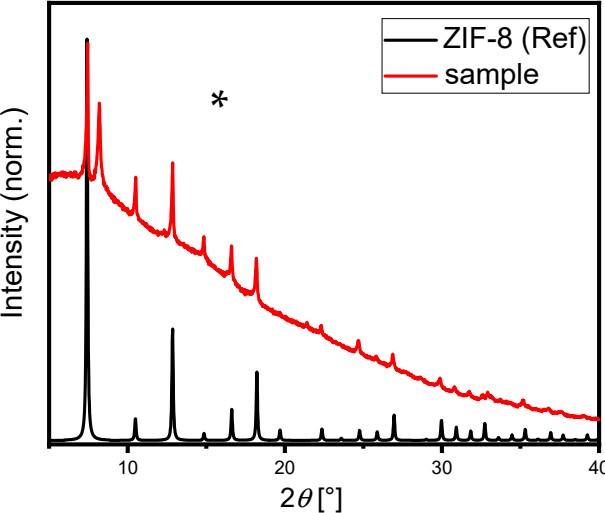

**Figure 3.** X-ray diffraction pattern of the deposited material (red) and corresponding ZIF-8 reference [35] (black), Cu–Kα1 radiation, λ = 1.54056 Å. The signal at 8.5°, marked with an asterisk, does not belong to the Bragg reflections of ZIF-8 material and corresponds to $Cu_{2.5}SO_4(OH)_3*2H_2O$ [36], which was formed as an admixture during anodic deposition of ZIF-8.

A selection of SEM images for specific conditions is shown in Figure 4. The Pictures 4B, 4E and 4H represent the standard synthesis conditions.

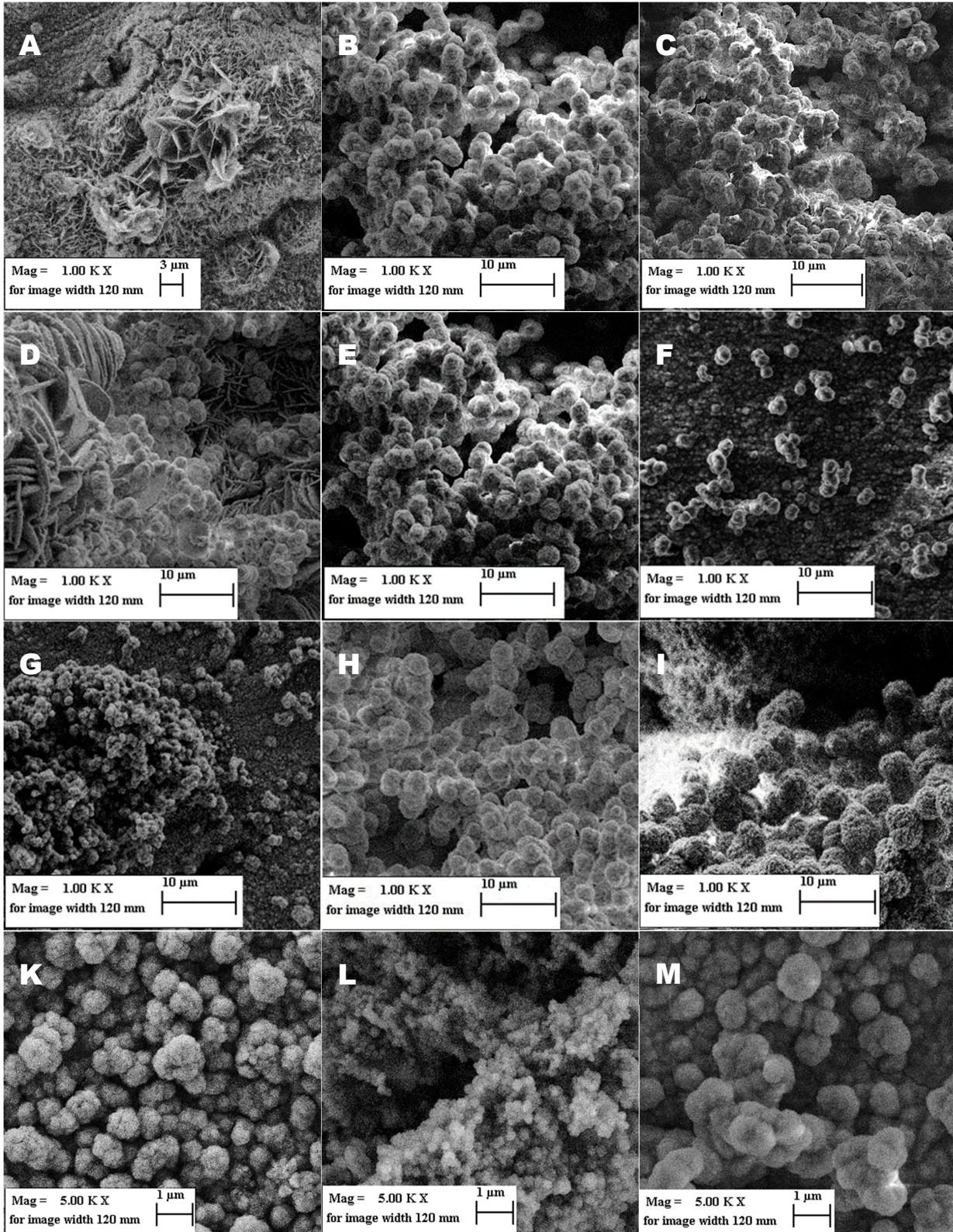

**Figure 4.** SEM images of ZIF-8 coating on a Cu substrate showing the impact of different synthesis conditions. (**A**–**C**) correspond to the varied linker concentration (with (**A**) = 0.5 M; (**B**) = 1.0 M; (**C**) = 2.0 M); (**D**–**F**) show the results of changing electrochemical potentials during the reaction (with (**D**) = −1.0 V; (**E**) = −1.1 V; (**F**) = −1.4 V); (**G**–**I**) demonstrate the effect of the reaction time (with (**G**) = 5 min; (**H**) = 30 min; (**I**) = 60 min), (**K**–**M**) represent the result of different electrolyte compositions ((**K**) = 25% EtOH, (**L**) = 50% EtOH, (**M**) = 75% EtOH).

### 3.1.1. Linker Concentration

Changing the concentration of the 2-methylimidazole-linker influences the nucleation and crystal growth process, independent from the dissolution process of zinc. Increasing the linker concentration reduces the residual amount of zinc in the electrolyte and accelerates the formation of the MOF. We tested concentrations of 2-methylimidazole with 0.1 M, 0.5 M, 1.0 M and 2.0 M as well as 1.376 M, as was reported in the literature [31]. For concentrations below 1.0 M, we could not observe a macroscopic deposition, while using SEM we could see a mix of flat intergrown and small spherical particles. Using concentrations above 1.0 M of 2-methylimidazole resulted in a coating which fully covered the exposed surface of the working electrode. X-ray diffraction studies showed diffraction patterns with Bragg reflections, which are characteristic for ZIF-8. The particle size was between 1.0 μm to 1.5 μm in diameter. The resulting samples for concentrations of 1.0 M and 1.376 M were nearly identical and showed well defined spherical particles, while a concentration of 2.0 M led to more intergrown particles. In order to achieve a better interaction of MOF and oil, we favor the spherical particles in contrast to the intergrown particles, due to the additional capillaries created. The SEM images for linker concentration 0.5 M, 1.0 M and 2.0 M are shown in Figure 4A–C. Generally, the deposited layer thickness increased with increasing linker-concentration, which had a negative impact on the mechanical stability of the layers. Mechanical stability of the coating was estimated during the rinse-off process, since rinsing the coating with water represents the main stress procedure for the sample during the synthesis and in subsequent applications.

### 3.1.2. Electrochemical Potential

By changing the potential between working and counter electrodes, the electromotive force is changed. A higher electromotive force leads to a higher current flow as well as a better distribution of zinc ions in the electrolyte. This generally accelerates the reaction itself, but also affects the crystal growth and the surface deposition, through higher dissolution rates of zinc. A higher dissolution rate leads to the formation of a high amount of smaller crystals [30]. The electrostatic force that increases with the increasing potential reduces the zinc concentration near the working electrode and forces crystal growth in the electrolyte, leading to more powder deposition than the desired layer deposition on the substrate. As shown in Figure 4D–F, a potential more positive than $-1.1$ V (vs. MSE) does not completely lead to the desired crystal shape. The deposited mass is increased, but the coating itself was mechanically unstable and contained the same flat intergrown particles, as in Figure 4A. Between $-1.1$ V and $-1.2$ V (vs. MSE), a layer deposition of ZIF-8 was observed Figure 4E), but with a potential more negative than $-1.2$ V (vs. MSE), the deposition increasingly changed to powder deposition. As seen in Figure 4F, a small number of scattered particles is deposited on the surface.

### 3.1.3. Synthesis Duration

An extended synthesis time affects the total amount of zinc dissolved, leading to extended crystal growth. The synthesis time for the deposition of a thinner layer must not fall below the critical limit needed for the incubation time [37]. That means that the nucleation period for the formation of crystal seeds can take more time than is provided for the reaction. Thus, not only the crystal size but also the number of crystals can be drastically reduced or fully inhibited. Variation of the synthesis time led either to a thin incomplete layer with small separate particles with <1 μm diameter for a synthesis duration of 5 min, or to well-defined particles with 1.0 μm to 1.5 μm diameter, creating a homogeneous complete coating for a synthesis time of 30 min, or spherical crystals with a size of 2.0 μm to 2.5 μm in diameter with a synthesis time of 60 min (see Figure 4G–I, respectively). The resulting coating for the longest synthesis time of 60 min showed a weak adhesion because of a high thickness. With an extended synthesis time, the particle size increases as well. Therefore, the optimal synthesis time must be as short as possible to create small particles, while

overcoming the incubation time of nucleation and creating a complete and homogeneous coating. For ZIF-8 synthesis, this corresponds to 30 min.

### 3.1.4. Electrolyte Composition

From the literature, it is known that the solvent plays an important role for electrochemical synthesis of MOFs in terms of solubility, ionic conductivity and deprotonation of the linker [30,32,38]. Thus, a lot of works studied the effect of different solvents onto the MOF formation, such as methanol and ethanol, especially for HKUST-1 [30,38]. Only very few works were devoted to investigations of ZIF-8 [38]. We tested the effects of ethanol concentrations in aqueous solutions onto ZIF-8 deposition. According to the theoretical considerations, a reduced water content decreases the conductivity and increases the nucleation rate [30], which should lead to an increasing number of smaller crystals. This would directly affect the layer thickness and would be beneficial for a thin (mono-)layer of electrodeposited ZIF-8. However, all experiments showed a poor particle growth for the ethanol concentrations of 25% (K), 50% (L) and 75% (M) shown in Figure 4. Due to the poor film quality, a more detailed characterization via XRD was not possible, the crystal size differs from unregular shaped particles with an average diameter of 1 μm (K and M) to more spherical shaped particles with a diameter of 200 nm. The poor deposition and yield are the result of the poor conductivity of the electrolyte and the poor solubility of the conductive salt. Therefore, there were no further investigations with other electrolyte additives.

### 3.2. Doctor Blade Technique for Composite Layer Containing PVDF Binder and ZIF-8

The doctor-blade technique is a very common method to create thin homogenous layers of powdered solids with defined thickness. This technique is often used, for example, for coatings of electrode materials for energy storage related applications [39]. A polymer is often added to the powder to increase adhesion and mechanical stability of the coating. By using polyvinylidene fluoride (PVDF), we chose a thermoplastic fluoropolymer that is itself known for its mechanical, chemical and relatively high thermal stability [40] as well as omniphobicity [41]. By combining porous ZIF-8 with PVDF, we created a mechanically stable coating of less than 25 μm thickness, which also has a high porosity and large surface area. As shown in Figure 5, the resulting coatings consist of small ZIF-8 particles, that are (due to ball milling) slightly smaller than the particles of the commercial ZIF-8 (Basolite® Z1200, Sigma Aldrich). The variation of the ZIF-8 to PVDF mass-ratio does not reflect significant differences in coatings morphology. The average ZIF-8 particles size is between 300–400 nm in diameter.

Further investigations via transmission electron microscopy (TEM) showed that ZIF-8/PVDF composite material obtained as a monolayer contains ZIF-8 particles embedded in a granular polymer matrix (Figure 6). EDS measurements of the selected region (Figure S1) show strong zinc signals consistent with ZIF-8. Due to the high sensitivity of the ZIF-8 material against the electron beam, more detailed investigations of crystallinity were not possible. However, XRD measurements of the monolayer scratched from the Cu substrate convincingly revealed the crystallinity of the ZIF-8 particles in the composite with PVDF, independent from the ZIF-8/PVDF ratio (Figure S1). Note that the two-layer successive coating is mechanically less stable in comparison to the monolayer, even under low mechanical stress. Therefore, the monolayer coating method was favored in the further investigation.

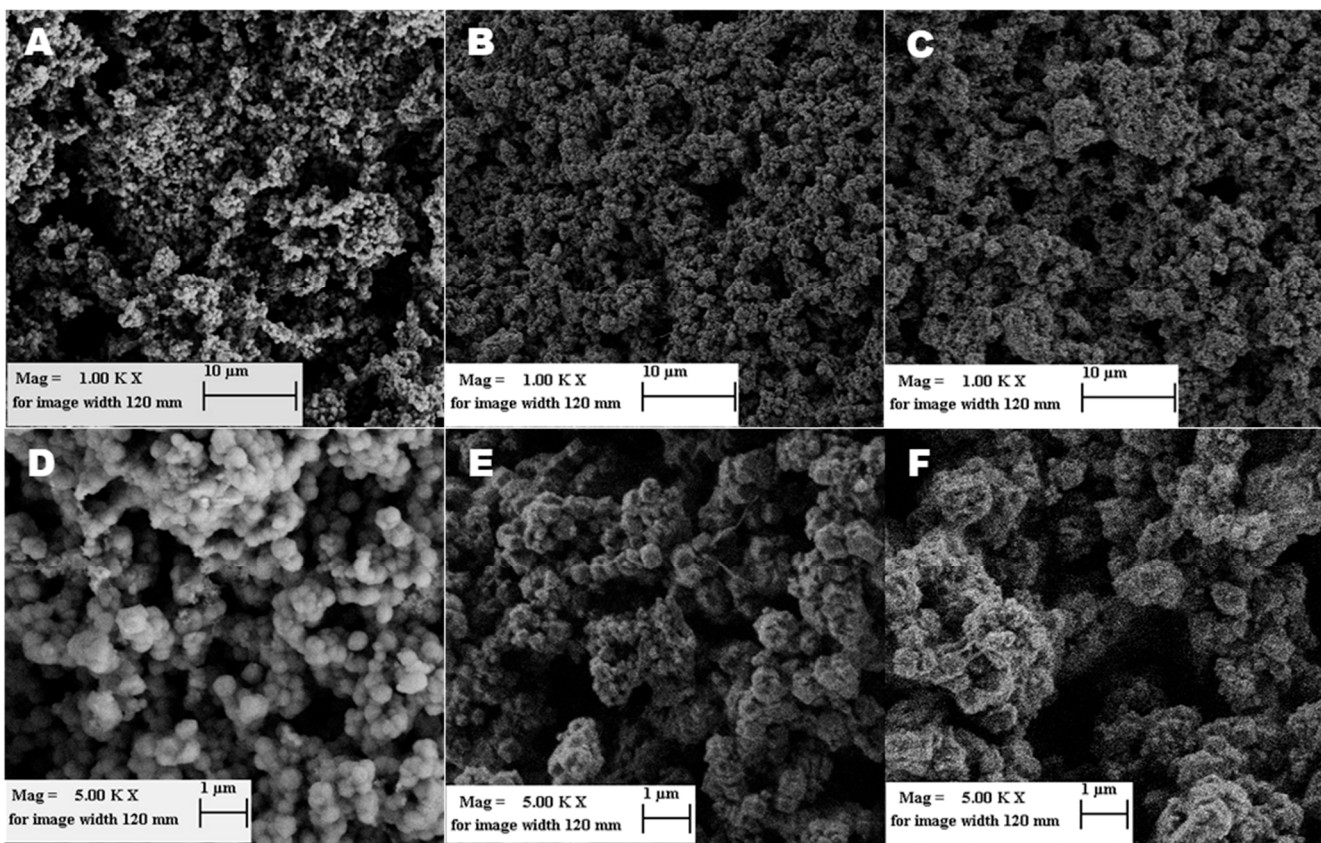

**Figure 5.** SEM images showing Basolite®Z1200 (**A**,**D**) and ZIF-8/PVDF composite layers with 10% (**B**,**E**) and 40% PVDF (**C**,**F**).

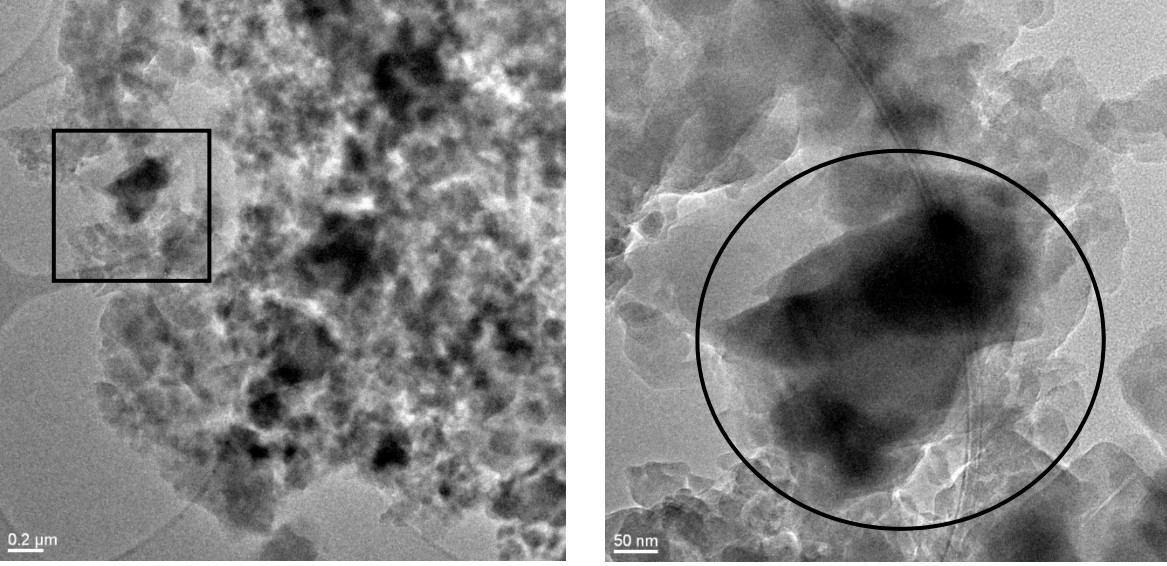

**Figure 6.** TEM images showing a ZIF-8/PVDF composite layer with 40% PVDF, at different magnification levels. EDS analysis of the selected area (right image) revealed a strong zinc signal, thus confirming the presence of ZIF-8 in the particle agglomeration (see Figure S2).

### 3.3. Surface Roughness Studies

The surface roughness of the resulting ZIF-8 layers after electrochemical synthesis and after coating with PVDF was measured using optical profilometry. Due to the limitation of the measuring device, the coatings were not infused with oil, as optically transparent media cannot be measured. The results are shown in Table 1. For a better comparison, the roughness of a pure PVDF-layer was also measured and compared with the roughness parameters of HKUST-1 [33], which was deposited using electrochemical anodization as well. The average roughness of three different ZIF-8/PVDF composite layers is three to five times lower than that of the corresponding reference sample (HKUST-1) and is close to the value of the pure PVDF. The electrochemically deposited ZIF-8 layer shows a value similar to the average roughness of the HKUST-1 layer, but has a significantly higher max peak. It can be assumed that subsequent oil infusion with GPL 105 or GPL 100, due to surface tension, will result in an additional reduction of the average roughness of all coatings.

**Table 1.** Surface roughness parameters—$R_a$: arithmetical average roughness, $R_q$: root mean square roughness, $R_p$: maximum peak height, $R_v$: maximum valley depth for ZIF-8 and PVDF layers. The parameters for HKUST-1 were taken from ref. [33] and included for comparison.

| Sample | $R_a$/μm | $R_q$/μm | $R_p$/μm | $R_v$/μm | Scan Length/mm |
|---|---|---|---|---|---|
| PVDF | 0.415 ± 0.00 | 0.522 ± 0.01 | 2.417 ± 0.06 | 1.953 ± 0.03 | 0.75 |
| ZIF-8/ PVDF (40%) | 0.895 ± 0.05 | 1.245 ± 0.09 | 6.561 ± 1.21 | 5.159 ± 0.34 | 0.75 |
| ZIF-8/ PVDF (20%) | 0.568 ± 0.04 | 0.780 ± 0.05 | 4.146 ± 0.35 | 3.130 ± 0.29 | 0.75 |
| ZIF-8/ PVDF (10%) | 0.510 ± 0.05 | 0.661 ± 0.07 | 5.655 ± 2.25 | 4.273 ± 1.09 | 0.75 |
| ZIF-8 (e-coating) | 3.411 ± 0.45 | 4.386 ± 0.63 | 61.990 ± 29.77 | 15.164 ± 2.99 | 0.75 |
| HKUST-1 [33] (e-coating) * | 2.233 ± 0.23 | 2.780 ± 0.26 | 11.097 ± 0.70 | 9.037 ± 0.70 | 1 |

* Roughness parameters were measured in one dimension instead of two dimensions for the ZIF-8 and PVDF layers.

### 3.4. Wetting Properties

The wetting properties of the ZIF-8 layers were assessed via contact angle measurements. The results are shown in Table 2. Measurements on a PVDF layer and on a HKUST-1 layer (values taken from [33]) are included for comparison. The results show that oil infusion with GPL 105 leads to a low contact angle hysteresis for both water and diiodomethane. Untreated ZIF-8 layers show high advancing contact angles, but the droplets are pinned to the surface so that the receding contact angles for water and diiodomethane could not be measured accurately. This is most likely due to the roughness of the highly porous ZIF-8 layers. Oil infusion with GPL 100 shows lower advancing contact angles than the untreated surfaces, but the droplets are also pinned to the surface. GPL 100 has a much lower viscosity than GPL 105 (20 cP vs. 800 cP at room temperature [42]). The higher viscosity of GPL 105 seems to result in a smoother oil surface, while after infusion with GPL 100, the surface roughness still has a considerable effect on the wetting properties. The variation of PVDF fraction in the ZIF-8 layers does not show any considerable effect on the advancing or receding contact angles. The wetting properties of the ZIF-8/PVDF (20%) monolayer are similar to the corresponding double layer. The ZIF-8 e-coating shows higher water contact angle hysteresis. This is most likely due to the higher roughness of the e-coating compared to the surfaces fabricated with the doctor-blade technique. Compared to the values of HKUST-1 taken from the literature [33], a considerably smaller contact angle hysteresis was achieved with the ZIF-8 layers. Considering all the contact measurements, the fluid properties of the oil and the roughness of the base surface structure are the key

parameters that define the wetting properties of the resulting oil-infused surface, while the surface chemistry of the base material has a little influence.

**Table 2.** Advancing contact angle $\theta_a$, receding contact angle $\theta_r$ and contact angle hysteresis $\Delta\theta$ of water and diiodomethane measured on ZIF-8 and PVDF layers. For receding contact angles reported as n/a, droplet pinning occurred and the contact line showed no significant movement. Values for oil-infused HKUST-1 are taken from ref. [33] and included for comparison.

| Surface | Oil | Water | | | Diiodomethane | | |
|---|---|---|---|---|---|---|---|
| | | $\theta_a$/deg | $\theta_r$/deg | $\Delta\theta$/deg | $\theta_a$/deg | $\theta_r$/deg | $\Delta\theta$/deg |
| ZIF-8/PVDF (20%) mono layer | none | 149.9 ± 4.0 | n/a | - | 127.8 ± 3.7 | n/a | - |
| | GPL 100 | 128.8 ± 4.0 | n/a | - | 97.1 ± 6.3 | n/a | - |
| | GPL 105 | 115.1 ± 4.3 | 104.1 ± 2.3 | 11.0 ± 4.9 | 97.4 ± 2.8 | 87.7 ± 2.7 | 9.7 ± 3.9 |
| ZIF-8/PVDF (20%) double layer | none | 149.2 ± 4.5 | n/a | - | 123.1 ± 6.9 | n/a | - |
| | GPL 100 | 130.9 ± 3.0 | n/a | - | 100.4 ± 4.9 | n/a | - |
| | GPL 105 | 115.6 ± 3.0 | 106.1 ± 2.6 | 9.5 ± 4.0 | 94.0 ± 5.3 | 86.4 ± 4.6 | 7.6 ± 7.0 |
| ZIF-8/PVDF (10%) | GPL 105 | 119.3 ± 1.2 | 112.6 ± 3.1 | 6.7 ± 3.3 | - | - | - |
| ZIF-8/PVDF (40%) | GPL 105 | 119.4 ± 2.4 | 111.6 ± 3.7 | 7.8 ± 4.4 | - | - | - |
| ZIF-8 (pellet) | GPL 105 | 116.5 ± 1.5 | 111.3 ± 2.0 | 5.2 ± 2.5 | 94.6 ± 1.4 | 87.9 ± 2.3 | 6.7 ± 2.7 |
| ZIF-8 (e-coating) | GPL 105 | 121.0 ± 4.9 | 98.0 ± 4.7 | 23.0 ± 6.8 | - | - | - |
| PVDF | none | 101.9 ± 2.2 | 41.9 ± 3.5 | 60.0 ± 4.1 | 82.2 ± 4.5 | 43.5 ± 3.7 | 38.7 ± 5.8 |
| HKUST-1 [33] (e-coating) | GPL 105 | 107.1 ± 1.3 | 5.7 ± 2.7 | 101.4 ± 3.0 | | | |

## 4. Discussion

In order to create a thin and homogeneous ZIF-8 layer as a host material in liquid infused porous surfaces (LIPS) using electrochemical anodization process, various synthesis conditions were evaluated. As the best conditions, a linker concentration of 1 M of 2-methylimidazole and 0.1 M of sodium sulfate as conductive salt were used. We applied a potential of −1.1 V (vs. MSE) over 20 min. The selected conditions resulted in a ~60-μm thin mechanically stable ZIF-8 layer, with an average ZIF-8 particle size of 1–1.5 μm. However, compared to the ZIF-8/PVDF composite layer, the electrochemically deposited ZIF-8 layer is still significantly thicker (min. 60 μm thickness) and has a higher roughness. This is mainly due to the larger particles, as the long incubation time did not allow a high nucleation rate, which would result in smaller particles and a simultaneous layer deposition. Additionally, material purity is an issue, as impurities due to crystallization from a solution are significantly greater than would be expected from crystallization from a melt [43]. In our case, the formation of Cu-containing admixtures arising from the Cu substrate reaction with the electrolyte additionally impedes the synthesis procedure. The usage of the blade technique has some advantage for the creation of the ZIF-8 layers, like an increased smoothness of the layer and a more precise control over the layer thickness.

The successive two-layer composite does not reveal any significant advantages in the apparent contact angles or contact angle hysteresis compared to the monolayer. Furthermore, the upper ZIF-8 layer can be detached from the polymer layer over time, so that due to this disadvantage, the monolayer coating was favored.

By comparing the contact angles of electrochemically created coatings with ZIF-8/PVDF composite layers and the ZIF-8 pellet, it can be noticed that the advancing contact angles (for GPL 105) have similar values around 118° for water and 96° for diiodomethane. The apparent (as placed) contact angles of these coatings are also similar to each other (apparent contact angles are shown in Table S1). However, the composite layers show a significantly lower water contact angle hysteresis of 9° compared to the electrochemically deposited samples with 23° hysteresis for ZIF-8. It seems clear that the surface chemistry and surface structure of the host layer has only little influence on the advancing contact angle, but a stronger influence on the resulting contact angle hysteresis. This can also

be seen from the comparison with the reported contact angle values of HKUST-1 [33]. Compared to the ZIF-8 surfaces, the different surface of the HKUST-1 base structure leads to a slightly lower advancing water contact angle of 107° but to a significantly higher water contact angle hysteresis of 101°. The advancing and apparent contact angles are mainly governed by the oil layer, while the contact angle hysteresis depends strongly on the properties of the base structure.

The underlying mechanism for this has not yet been conclusively clarified, but in our opinion, there are two possible effects can explain these findings. It has been shown before that stable oil infusion is achieved by hemiwicking, but surface features of the underlying host layer can penetrate the oil and are then in direct contact with the droplet on top [33]. This creates an inhomogeneous surface that is characterized by a different surface chemistry and morphology as well as different surface tensions. This inhomogeneity leads to contact line pinning [44–46], which increases the contact angle hysteresis. By adding PVDF to the surface coating, the differences in polarity and surface tension between oil and penetrating spikes can be reduced, leading to a more chemically homogeneous surface. At the same time, the interaction between MOF/PVDF composite and the perfluorated oil is increased, which leads to a better oil infusion of the spikes. This effect can explain the lower contact angle hysteresis on the ZIF-8/PVDF composite layers compared to the electrochemically deposited ZIF-8 samples. The fact that we see no dependence between hysteresis and PVDF content, as well as the fact that we did not see any PVDF-enriched spikes in TEM studies, indicates a second effect that leads to the differences in the contact angle hysteresis. This second effect is directly related to the surface roughness. A smoother host material decreases the amount and size of spikes penetrating the oil. This leads to a more chemically and morphologically homogeneous surface, and therefore, reduces contact line pinning. This effect can be the reason for the lower water contact angle hysteresis on the composite layers, which corresponds to a lower roughness compared to the electrochemically deposited ZIF-8 samples and the HKUST-1 coating.

## 5. Conclusions

In this work, we optimized the electrochemical deposition of ZIF-8 films to use them as host materials for LIS. Furthermore, alternative ZIF-8-based host composite layers were developed and characterized. As expected from the literature, the comparison of the contact angles of the LIS layers showed that the resulting contact angles are largely independent of the host material and are significantly influenced by the infusion agent (Krytox GPL 105). However, the contact angle hysteresis is influenced by the host material and shows a significant reduction for smoother host layers in contact angle measurements with water.

The electrodeposited ZIF-8 layers showed a significantly increased roughness and a contact angle hysteresis, due to larger ZIF-8 particles. The smoothness of the ZIF-8 coating can further be improved by achieving smaller particle diameters, although this requires more extensive optimization of the synthesis conditions in order to significantly shorten the incubation time, which is a significant limiting factor. The coatings produced show good contact angles and contact angle hysteresis as low as 6.7°. They are, therefore, well suited for industrial applications in the context of dropwise condensation. To further investigate the wetting effects of the oil-infused coatings, advanced methods of contact angle analysis [47] can be helpful in future research. Furthermore, the coating methods without oil infusion can be used to deposit thin MOF layers on metallic surfaces.

**Supplementary Materials:** The following are available online at https://www.mdpi.com/article/10.3390/app11094041/s1. **Figure S1**. X-ray diffraction pattern of the ZIF-8-PVDF composite monolayer. **Figure S2.** TEM imaging with EDS spectrum. **Table S1.** Static contact angles.

**Author Contributions:** Correspondence, J.S. and M.S.; TEM measurements, I.G.G.M.; project administration, D.M. and S.U.; supervision, S.K. All authors have read and agreed to the published version of the manuscript.

**Funding:** This research was funded by the European Union and the Free State of Saxony under the TroKo project (SAB Grant No. 100317611/100317620).

**Institutional Review Board Statement:** Not applicable.

**Informed Consent Statement:** Not applicable.

**Data Availability Statement:** Not applicable.

**Acknowledgments:** We would like to acknowledge Volker Hoffmann (IFW Dresden) for assistance with profilometry measurements. The publicaiton of this article was funded by Open Access Funding by the Publication Fund of the TU Dresden.

**Conflicts of Interest:** The authors declare no conflict of interest.

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
