# Peer review of "Preparation and Application of ZIF-8 Thin Layers"

_applsci, doi:10.3390/app11094041_

Round 1

Reviewer 1 Report

The authors have shown a novel and original work. The points to be amended are mentioned below:

(1) It would be useful to mention the purity levels of all chemicals used in the experimental methods section.

(2) In the first sentence of Section 3.1.4 as it is mentioned "from the literature" the related references should be mentioned at the end of the sentence.

Thank you.

Author Response

Dear Reviewer

We will send the notes and answers for correction as a pdf.The corrected version of the manuscript and the SI are combined in a .zip file both with marked changes (green) with the name suffix "..._marked" and without colored changes (no name suffix).  The newly inserted images are also included as original files.
Since the suggestions of all reviewers have led to a change in the manuscript, we have combined both comments in the pdf to create a better overview.
We hope for your understanding and would like to thank you again for your comments.

Yours sincerely

The authors

Reviewer 2 Report

Below you will find my impressions concerning the manuscript Preparation and application of ZIF-8 thin layers from Schernikau, Sablowski et al..

From reviewers opinion the work is a very interesting well written contribution. From reviewer’s opinion the article does need moderate revisions before publication.

I suggest to improve the CA analysis procedure ((experimental) in the future) using state-of-the-art methods and try to present the work in a more general surrounding (e.g. lubricant impregnated surfaces, pinning, …)

Revisions

1.    Line 35, 105: Please clarify high CA and low CA hysteresis (large tha xx deg, samller than yy deg).

2.    A Scheme of the MOF Structures and some of the synthesis procedures or of the used synthesis procedure might be interesting for the article.

3.    Doctor plate method: Please at the area of the coated film (substrate size). How much of the mixture was placed on the substrate? Please clarify the procedure. Amount/size/kind of balls, who did you remove the balls,…?

4.    Please describe the CA procedure, volume, volume change, fitting procedure Ellipse, Laplace (?)…. E.g.  “No receding angle is …”- Clarify if the droplets are pinned, compare to figures in the references (Schmitt and Heib HPDSA, Bormashenko, K. Mittal…). Note that axisymmetric analysis procedures of nearly all companies cannot analyse such CAs. I suggest to improve your procedure (in the future). A simple cover and some delay times helps to make the experimental measurements more reliable.  

o   You mentioned static contact angle but dynamically analysed the contact angle? The “(equilibrium)” defiantly is not true and cannot stand. Please explain. Please use a consistent concept for CA definitions and description. I found a heat transfer publication (heat transfer and fouling setup subboiling) contains an alternated concept of H. Kamusewitz, (W. Possart and D. Paul, Colloids and Surfaces a-Physicochemical and Engineering Aspects, 1999, 156, 271-279.) to obtain equilibrium CA from contact angle hysteresis. Correlation with YOUNG equation is only possible for a thermodynamic equilibrium angle (with your method it is questionable to observe the equilibrium angle especially for DCM). A number of researchers are convinced that this is only the case for vibrating experiments or if the drop is large (concept of most stable CA “Marmur” ), others are convinced that the angle is not easily obtainable due to the macroscopic observation of the tri-phase interface (optical) and the activation energy which is necessary to move/form the triple line. But you do not want to solve the problem of correct CA measurements. Hence cite some references dealing with the problem of correct/possible definition, measurement or analysis (L. F. M. da Silva, A. Oochsner, R. D. Adams, M. E. R. Shanahan and W. Possart, Springer-Verlag Berlin Heidelberg, 2011, ch. 4.; J. Drelich, Surface Innovations, 2013, 7.; E. Bormashenko, Colloid. Polym. Sci., 2013, 291, 339-342. M. Schmitt, K. Gross, J. Grub and F. Heib, J. Colloid Interface Sci., 2015, 447, 229-239. )   sorry that I digressed, call the static contact angles simply “apparent contact angle”, I guess it is the contact angle after placing the droplet onto the surface?

o   You might check on the publications dealing with slippery surfaces / lubricant impregnated surfaces.

o   The pinning surface might be interesting to be analysed by statistical procedures firstly presented by Heib and Schmitt.

o   You might use the word pinning (more often) within the discussion. The 101° hysteresis might also be influenced by the drop size (influence of the needle, you can find contributions of the above mentioned authors)

5.    Figure 2 I would exchange the word “reflexion” by “signal”.

6.    An image of the electrochemical setup(s), e.g within the supporting information might improve the value of the manuscript.

7.    How did you check the mechanical stability? If not, do not state this as a fact.

8.    Discussion: The part concerning the pinning are not convincing (455)? Why should the presence of surface structures which the oil has a large effect on the advancing angle and the apparent contact angle (which is the advancing angle of the drop placing). Some ideas of force effects of dynamic contact angles on horizontal and inclined surfaces can be found in literature.

9.    Line 470 I suggest sorting the example the other way around which is more conclusive with the discussion. You started with large contact hysteresis due to large number of pins (spikes) to no/small contact angle hysteresis for the smooth surface. But for the example you wrote “… can be the reason for the higher contact angle hysteresis”

Author Response

Dear reviewer

We will send the notes and answers for correction as a pdf.The corrected version of the manuscript and the SI are combined in a .zip file both with marked changes (green) with the name suffix "..._marked" and without colored changes (no name suffix).  The newly inserted images are also included as original files.
Since the suggestions of all reviewers have led to a change in the manuscript, we have combined both comments in the pdf to create a better overview.
We hope for your understanding and would like to thank you again for your comments.

Yours sincerely

The authors
